

# Effects of Temperature, Pressure, and Carrier Gases on the Performance of an Aerosol Particle Mass Analyser

Ta-Chih Hsiao[1], Li-Hao Young[2], Yu-Chun Tai[1], Po-Kai Chang[1]

[1]Graduate Institute of Environmental Engineering, National Central University, Taoyuan, 32001, Taiwan

[2]Department of Occupational Safety and Health, China Medical University, Taichung, 40402, Taiwan

*Correspondence to*: Ta-Chih Hsiao (@ncu.edu.tw)

**Abstract.** Effective density is a crucial parameter used to predict the transport behaviour and fate of particles in the atmosphere, and for measuring instruments used ultimately in the human respiratory tract (Ristimäki et al., 2002). The aerosol particle mass analyser (APM) was first proposed by Ehara et al. (1996) and is used to determine the effective density of aerosol particles. A

compact design (Kanomax APM-3601) was subsequently developed by Tajima et al. (2013). Recently, a growing number of field studies have reported application of the APM, and experimental schemes using the differential mobility analyser alongside the APM have been adopted extensively. However, environmental conditions such as ambient pressure and temperature vary with the experimental location, and this could affect the performance of the APM. Gas viscosity and Cunningham slip factors are parameters associated with temperature and pressure and are included in the APM's classification

performance parameter: $\lambda$. In this study, the transfer function and APM operational region were calculated and discussed to examine their applicability. Air, oxygen, and carbon dioxide were selected to atomize aerosols in the laboratory with the aim of evaluating the effect of gas viscosity on the APM's performance. Using monodisperse polystyrene latex spheres with nominal diameters of 50 and 100 nm, the experimental results for the classification accuracy of the APM were consistently within 10%. Based on the theoretical analysis of the APM's operational region, the lowest mass detection limit can be extended

from $5.9 \times 10^{-2}$ to $1.4 \times 10^{-3}$ fg when the carrier gas is hydrogen (viscosity = $8.82 \times 10^{-4}$ N·s/m²) with a chosen $\lambda$ value of 0.5. Moreover, it can be further extended to $7.0 \times 10^{-4}$ fg when the pressure is reduced from 101.3 to 80 kPa, which implies that performance may be affected during field study. These results provide an insight into the ability to extend the classifiable size range without modifying the hardware of the APM or scarifying the classification resolution.

## 1 Introduction

To determine the adverse health effects of inhalable particles, the lung regional deposition fraction must be investigated (Chuang et al., 2016; Haddrell et al., 2015). The lung regional deposition fraction is closely associated with the density and morphology of submicron particles that can be derived from their mass measurement (Broday and Rosenzweig, 2011; Salma et al., 2002; Shi et al., 2015). Therefore, mass distribution and particle density play pivotal roles in the study of associated health effects. However, obtaining measurements of the morphology or density of aerosol particles in the environment is not



easy (Bau et al., 2014; DeCarlo et al., 2004), and particles in an ambient environment are generally irregular and nonspherical. In this regard, Liu et al. (2013) demonstrated that the morphological parameter of a fractal soot particle, namely the fractal dimension ($D_f$), could affect the particle's radiative properties. In addition, according to theoretical calculations, He et al. (2015) reported that aged soot aggregates with partially encapsulated or externally attached structures have weaker absorption properties than do fresh soot aggregates.

Nevertheless, theoretical simulations and experimental studies on the effects of aerosols on human health and the atmosphere all require certain assumptions regarding particle morphology or density to be made to enable the conversion of number concentration to mass concentration or volume concentration (Hand and Kreidenweis, 2002). To retrieve the density of submicron-to-nanometre-sized particles, a differential mobility analyser (DMA) system coupled with a low-pressure impactor was developed to measure the aerodynamic sizes of particles of known electric mobility size. By further assuming that the particles were all spherical, the "effective" or "apparent" density was derived (Kelly and McMurry, 1992; Schleicher et al., 1995; Skillas et al., 1998). A similar system combining a scanning mobility particle sizer and electrical low-pressure impactor in parallel was reported and applied to study diesel particulate matter and ambient aerosols (Maricq et al., 2000; Ristimäki et al., 2002; Symonds et al., 2007; Van Gulijk et al., 2004; Virtanen et al., 2004). Furthermore, (Ehara et al., 1996) developed the aerosol particle mass analyser (APM), which is an aerosol instrument that classifies particle mass by balancing the centrifugal force and electrostatic force. Based on an identical classification mechanism but utilizing a different rotating scheme to create centrifugal force, the Couette centrifugal particle mass analyser is another commercially available instrument. With such advances in aerosol instrumentation, McMurray et al. (2002) proposed using the APM to directly measure the mass of monodispersed particles classified by a DMA (tandem DMA–APM system) instead of using an impactor to probe the particle aerodynamic size. This DMA–APM scheme is capable of revealing the density or mass distribution of targeted aerosol particles in real time. Throughout the past decade, this scheme has also been adopted extensively to determine the $D_f$ of aerosol aggregates (Lall et al., 2008; McMurry et al., 2002; Park et al., 2003; Park et al., 2004a; Park et al., 2004b; Scheckman et al., 2009).

A growing number of APM experiments are being conducted in outdoor environments in relation to the dual interest in human health and climate change (Leskinen et al., 2014; Rissler et al., 2014). However, when experiments are conducted in the field, environmental conditions such as temperature ($T$) and pressure ($P$) vary spatially and temporally, particularly at high-altitude sites. The viscosity ($\mu = \mu_r \left(\frac{T_r+S_u}{T+S_u}\right)\left(\frac{T}{T_r}\right)^{1.5}$) and mean free path ($\ell = \frac{\mu}{P}\sqrt{\frac{\pi RT}{2M}} = \ell_r \left(\frac{P_r}{P}\right)\left(\frac{T}{T_r}\right)^2 \left(\frac{T_r+S_u}{T+S_u}\right)$) of ambient air change continuously (Kulkarni et al., 2011), thereby influencing the performance and range of detection limits (classifiable region) for the APM. Therefore, in this study, air, oxygen, and carbon dioxide were selected as carrier gases to evaluate the effect of gas viscosity and the mean free path on the performance of the APM, including the classifiable region and detection limits. Argon would be required as the carrier gas if the APM was used as an aerosol particle classifier coupled with inductively





coupled plasma mass spectrometry (ICP-MS; in a similar manner to the DMA–ICP-MS system) (Myojo et al., 2002). Therefore, the effects of changing the carrier gas in the APM's transfer function require investigation. In this regard, the experimental and simulated transfer functions for the APM operated under various carrier gases were explored and are compared in this paper. The results provide a valuable insight into the performance of the APM when operated under various

conditions.

## 2 Operational theory

The APM was first proposed by Ehara et al. (1996) and a compact version was recently developed by Tajima et al. (2013). The instrument consists of two electrodes, namely coaxial and rotating cylindrical electrodes, between which a narrow annular space is created for mass classification (classification zone). When the APM is initialized, the inner and outer electrodes rotate

at the same speed ($\omega$) to generate centrifugal force, and a high voltage is applied to the inner electrode to create a "counter" electrostatic force. The governing equations of particle movements in radial and axial directions inside the classification zone are expressed as follows (Ehara et al., 1996):

$$\frac{m}{\tau} \cdot \frac{dr}{dt} = m \cdot r \cdot \omega^2 - \frac{q \cdot V}{r \cdot \ln(r_o/r_i)} \quad \text{(in a radial direction),} \tag{1}$$

$$\frac{m}{\tau} \left\{ \frac{dz}{dt} - v(r) \right\} = 0 \quad \text{(in an axial direction),} \tag{2}$$

$$\tau = \frac{\rho_p \cdot d_{p,m}^2 \cdot C_c(d_{p,m})}{18\mu}, \tag{3}$$

$$S_c = \frac{m}{q} = \frac{V}{r_c^2 \cdot \omega^2 \cdot \ln(r_0/r_i)}. \tag{4}$$

Particles are then classified based on the balance between the centrifugal force and counter electrostatic force, and only particles with a designated mass-to-charge ratio ($S$) pass through the classification zone. The mass-to-charge ratio for particles that remain in the central line ($r_c$) within the classification zone is defined as the "critical" $S$ ($S_c$; Eq. (4)). Based on this

definition, the classified particle mass, rotation speed, and voltage applied are linearly correlated on a log scale, as shown in Eq. (5):

$$\log(m) = \left[\log(q) - 2\log(r_c) - \log\left(\ln(r_0/r_i)\right)\right] + \log(V) - 2\log(\omega) \tag{5}$$

Tajima et al. (2011) and Tajima et al. (2013) depicted the classifiable region for the APM by using a log–log mass versus rotation speed plot. However, because of the physical limits of the operational voltage and rotation speed of the APM, a parallelogram of potential classifiable regions was reported.

The transfer function ($\Omega$) is generally used to characterize the classification performance of the DMA, and by defining the

penetration probability of particles under a designed condition, the transfer function is employed to evaluate the performance



of the APM. The nondiffusive transfer function for the APM ($\Omega_{APM}$) can be indexed by the penetration at the $S_c$ ($t(S_c)$) and the resolution parameter ($\Delta S$) (Tajima et al., 2013); $t(S_c)$ is the maximum height of the transfer function and $\Delta S$ can be estimated theoretically as follows:

$$t(S_c) = exp(-\lambda_c),\tag{6}$$

$$\Delta S = \frac{r_c}{4\delta} tanh(\lambda_c/2).\tag{7}$$

Both parameters are closely related to $\lambda_c$, which is the major dimensionless performance parameter defined by(Ehara et al., 1996) for APM operation. The $\Omega_{APM}$ is a strong function of $\lambda_c$:

$$\lambda_c = \frac{2\tau \cdot \omega^2 \cdot L}{\bar{v}} = \frac{2 \cdot \frac{\rho_p \cdot d_{p,m}^2}{18} \cdot \omega^2 \cdot L}{\left(Q_a \middle/ \pi(r_o^2 - r_i^2)\right)} \cdot \frac{C_c(d_{p,m})}{\mu} = \frac{2m/3d_{p,m} \cdot \omega^2 \cdot L}{\left(Q_a \middle/ (r_o^2 - r_i^2)\right)} \cdot \frac{C_c(d_{p,m})}{\mu}.\tag{8}$$

$\lambda_c$ can be interpreted as the ratio of the time scale for axial and radial particle movements in the classification zone. When the

value of $\lambda_c$ is larger, the APM has a superior classification resolution. Generally, the axial traversal time scale ($L/\bar{v}$) is considered constant because the aerosol flowrate is fixed when the instrument is operated. Thus, the classification performance can be improved by decreasing the radial traversal time scale ($1/2\tau \cdot \omega^2$). A shorter radial traversal time enables easier removal of incompetent particles by deposition onto the electrodes, which denotes lower penetration possibility for particles with a mass-to-charge ratio other than $S_c$. In other words, the transfer function is narrower, or $\Delta S$ is smaller, for a larger value of $\lambda_c$,

which suggests a higher classifying resolution. However, the trade-off for operating the APM at a larger $\lambda_c$ is that the maximum penetration of the transfer function ($t(S_c)$) is lower. Tajima et al. (2011) recommended that the APM be operated at a constant $\lambda_c$ (with a fixed $\Omega_{APM}$) within the range of 0.25 to 0.5 when the aerosol flow rate is 0.3 lpm.

For a constant value of $\lambda_c$, a unique curve in the log–log mass versus rotation speed plot can be determined iteratively through

Eq. (8), and this intercepts the APM classifiable region. This constant-$\lambda_c$ curve generally acts as the lower boundary for the APM classifiable region. As observed in Eq. (8), $\lambda_c$ is a function of the APM rotation speed ($\omega$) and depends on particle mass and density with respect to relaxation time ($\tau$). The value of $\tau$ is closely related to gas viscosity ($\mu$) and the Cunningham slip factor ($C_c$). At a given $\lambda_c$, the $\omega$ for classifying particles with a certain mass decreases with an increasing $C_c$-to-$\mu$ ratio ($\frac{C_c}{\mu}$). For air, oxygen, argon, and carbon dioxide, limited variations in the values of $\frac{C_c}{\mu}$ are observed, and their constant-$\lambda_c$ curves are

closely clustered. By contrast, the long mean free paths of low molecular weight gases such as hydrogen and helium can lead to a relatively high $\frac{C_c}{\mu}$. Consequently, when the APM is operated with hydrogen and helium, the classifiable mass range needs to be extended to a much lower detection limit without sacrificing the resolution or modifying the APM hardware (Fig. 1). However, an implicit problem is that the breakdown voltages of inert gases are generally approximately one order of magnitude lower than that of air (Schmid et al., 2002) (values of gas-specific parameters and the corresponding detection limits are listed

in Table 1).





The expression of $\frac{C_c}{\mu}$ can be further rewritten to reveal the effects of temperature, pressure, and the carrier gas species as Eq. (9):

$$\frac{C_c(d_p)}{\mu} = \frac{1}{\mu} + \frac{2\ell/d_p}{\mu}\left[\alpha + \beta \cdot exp\left(\frac{-\gamma}{2\ell/d_p}\right)\right] = \left(\frac{T+S_u}{T_r+S_u}\right)\left(\frac{T}{T_r}\right)^{-1.5} \cdot \frac{1}{\mu_r} + \left(\frac{P_r}{P}\right)\left(\frac{T}{T_r}\right)^{0.5} \cdot \frac{2\ell_r/d_p}{\mu_r}\left[\alpha + \beta \cdot exp\left(\left(\frac{P}{P_r}\right)\left(\frac{T}{T_r}\right)^{-2}\left(\frac{T+S_u}{T_r+S_u}\right) \cdot \right.\right.$$

$$\left.\left.\left(\frac{-\gamma}{2\ell_r/d_p}\right)\right)\right], \tag{9}$$

where $\alpha = 1.142$, $\beta = 0.558$, and $\gamma = 0.999$ (Allen and Raabe, 1985). As shown in Fig. 2, compared to pressure, $\frac{C_c}{\mu}$ is somewhat

invariant with temperature; when the temperature changes from 273 to 318 K, the difference of $\frac{C_c}{\mu}$ for all gas species is less

than 6%. However, the $\frac{C_c}{\mu}$ for 100-nm particles increases by 42% when the air pressure decreases to 65 kPa (Fig. 3); therefore,

when operating the APM at high-altitude sites, the detection limits need to be lowered or the resolution needs to be improved.

By contrast, based on the analytical predictions, Eq. (4) is unaffected by $\frac{C_c}{\mu}$, and the classification accuracy remains unchanged.

## 3 Experimental method

To experimentally evaluate the classification accuracy, polystyrene latex (PSL) spheres certified by National Institute of Standards and Technology were classified using the DMA (TSI 3081) and then delivered to the APM (Kanomax model Ⅱ - 3601) to determine particle mass. The overall aerosol flow rate of the DMA–APM system was controlled by the downstream CPC (Condensation Particle Counter, TSI 3022A) and fixed at 0.3 lpm. To optimize the size of the classification resolution, the sheath flow rate of the DMA was set at 3.0 lpm. In addition to air, carbon dioxide ($CO_2$) and oxygen ($O_2$) with purity levels of 99.99% were supplied as carrier gases. Before conducting measurements, the selected gas was used to purge all apparatus for at least 5 min to ensure that no residual contaminant gases remained, and the flowrates of the DMA's sheath flow and aerosol flow were calibrated using a volumetric flow meter (Gilian Gilibrator 2, Sensidyne, St. Petersburg, FL, USA). Because the DMA–APM was used to investigate the effects of carrier gases on the APM's performance, it was necessary to first study the sizing accuracy of the DMA under various carrier gases. The results showed that the differences between the measured modal diameters and nominal sizes of 50- and 100-nm certified PSL particles were within 6% for air, $CO_2$ and $O_2$. Therefore, no significant effects of gas species on DMA sizing accuracy were observed, which was consistent with the findings of Schmid et al. (2002).

The classifying accuracy of aerosol instruments under various conditions is generally characterized by a normalized indicator relative to a known reference (Karg et al., 1992; Marlow et al., 1976; Ogren, 1980; Schmid et al., 2002). The normalized mass-to-charge ratio ($\bar{S} = \frac{S_{gas}}{S_{air}}$) was used in the present study to analyse the APM's performance under a constant rotation speed ($\omega$) and constant sizing resolution ($\lambda$) operation. For the constant $\lambda$ operation, $\lambda$ values of 0.24 and 0.45 were chosen for both 50-





and 100-nm PSL sphere. The APM was operated with step-ramp voltages, and the corresponding particle concentrations at the outlet were measured by the CPC. Typical APM $C_N$-V spectra were inspected.

Detailed studies on the APM's $C_N$-V spectra were conducted by investigating the transfer function through software simulation.

The transfer function is a kernel function used to theoretically evaluate the concentration of downstream particles, and is critical for evaluating the performance of the APM (Ehara et al., 1996; Emery, 2005; Tajima et al., 2013). In the present study, two computer programs, namely the TRANSFER program and SIM_APM program (developed by the National Institute of Advanced Science and Technology [AIST] of Japan) were utilized to theoretically evaluate the operational performance of the APM (Tajima et al., 2011; Tajima et al., 2013). The TRANSFER program calculated the theoretical APM transfer function at

a fixed rotation speed and voltage, and the SIM_APM program simulated particle distribution at the APM outlet based on the known particle distribution at the APM inlet (the size distribution was classified by the front DMA). Both simulated results were compared with experimental APM measurements.

## 4 Results and discussion

### 4.1 Constant λ

For the constant λ operation case, $\bar{S}$ is expressed as Eq. (10), where $\bar{S}$ is mainly a function of the voltage applied to the APM ($V$), Cunningham slip correction factor ($C_c$), and gas viscosity ($\mu$):

$$\bar{S} = \frac{S_{gas}}{S_{air}} = \frac{V_{gas}}{V_{air}} \frac{C_{c,gas}/\mu_{gas}}{C_{c,air}/\mu_{air}}. \tag{10}$$

To consider the various properties of the gas species, $\omega$ was adjusted to enable λ to be fixed at approximately 0.24 and 0.45 for 50-nm PSL particles, as shown in Table 2. The results revealed that particle mass was generally underestimated for cases

where $CO_2$ was used as a carrier gas. In particular, underestimation was 23%–25% for a 50-nm PSL sphere. By contrast, when $O_2$ was used as the carrier gas, an overestimation of mass measurements was observed, with an error within 9%.

To further eliminate the bias propagated from DMA classification, the transfer function of the APM was calculated using software developed by the AIST of Japan. The transfer function predicted (hereafter referred to as "predicted $\Omega_{APM}$") based on

the known size distribution of the DMA outlet (convoluted with the known size distribution classified by DMA) is indicated by the bold purple line in Fig. 6, and the theoretical transfer function (hereafter referred to as "theoretical $\Omega_{APM}$") at a fixed optimal experimental peak voltage is indicate by a thinner blue line. Experimental data points are indicated by the dotted symbol in Fig. 6.

The APM resolution defined as $s_c/\Delta s$ is used as an indicator to evaluate the classification performance. The resolution is recommended to be between 0.5 and 1.2 when interpreted as $\Delta s/s_c$ (Emery, 2005; Tajima et al., 2011). However, in this study,





the values of $s_c/\Delta s$ for $CO_2$ were 0.43 and 0.44 for 50- and 100-nm PSL spheres, respectively. Furthermore, the experimental optimal voltage in the case of $CO_2$ was consistently lower than the theoretical voltage after convolution with the classified size distribution, and slightly higher than the theoretical voltage in the case of $O_2$. As shown in Table 1, the viscosity of $CO_2$ was lower than that of air, whereas the viscosity of $O_2$ was higher than that of air. These findings exhibit qualitative agreement

with observations of under- or over-estimations of PSL spheres. Therefore, we suspect that the fluid field in the APM classification zone, also known as Taylor–Couette flow, is influenced by gas-specific properties such as $\mu$ and $\rho$. A further numerical simulation of the flow field was performed using COMSOL Multiphysics 4.3a. Using the flow velocity of air as a reference, the velocity differences in an angular direction at the APM's inlet and outlet under various $\omega$ values are plotted in Fig. 7. The velocity was generally lower in the classification zone of the APM when $CO_2$ was used as the carrier gas, and an

increase in the distinct differences between $CO_2$ and air with an increase in the rotation speed was observed. Therefore, a lower viscosity and higher gas density likely intensify the shear force required to create rotating flow inside the APM. Because of the lower rotating flow velocity, significant deviations were observed in the measured results under normal conditions in the case of $CO_2$; this phenomena is intensified with higher values of $\omega$ and is more significant for small particles, which are even more prone to influence from the flow field.

**4.2 Constant ω**

As shown in Table 3, when determining the particle mass for 100-nm PSL spheres, respective differences of -14% and +6% are observed when $CO_2$ and $O_2$ are used as carrier gases in the APM. However, in Eq. (4), $S_c$ is independent of gas properties, and thus $S_c$ or the peak voltage of the $C_N$-V spectrum should remain unchanged if identical values of $\omega$ are applied in the APM under various carrier gases. The current experimental results demonstrate that the classification ability of APM is dependent

on the carrier gas and influenced by gas viscosity. There could be changes in the flow field within the APM when $CO_2$ and $O_2$ are used, and radial acceleration may not be $r_c\omega^2$, as used in Eq. (4). Therefore, to determine the exact centrifugal force for $S_c$, further research is required to investigate the velocity profile of particles in an angular direction. In addition, when comparing air and $O_2$, a broader $C_N$-V spectrum with weaker penetration was observed for $CO_2$; this could be attributed to the higher diffusivity resulting from the lower gas viscosity of $CO_2$ (Stokes–Einstein equation: $\mathcal{D} = k_B T / 3\pi\mu d_p$). Based on the results,

on-site calibration of the APM's classification performance is strongly recommended.

**5 Conclusion**

In this study, the effects of temperature ("T" hereafter), pressure ("P" hereafter), and gas viscosity ("μ" hereafter) on the performance of the APM were evaluated analytically and experimentally. The analytical results revealed that the APM's detection limit can be lowered simply by increasing the Cc-μ ratio without modifying the hardware of the APM or its

classifying resolution. Under a constant λ and fixed T and P, the use of a low molecular weight carrier gas such as $H_2$ or He can lower the mass detection limit by an approximate order of $<10^{-2}$ fg. Similarly, a reduction in operating P lowers the



detection limit or improves the resolution. Under these circumstances, the effects of T on the APM's detection limit are relatively minor. Our experimental results under constant $\lambda$ or $\omega$ values reveal that the use of a carrier gas other than air reduces accuracy. Specifically, a carrier gas with a lower $\mu$ than air such as $CO_2$ yields an underestimation of mass, whereas one with a higher $\mu$ such as $O_2$ yields an overestimation. A subsequent flow field simulation revealed variations in flow velocity in an angular direction at the APM's inlet and outlet when air and other carrier gases were used. The flow velocity decreased with $\mu$ but increased with $\omega$. Thus, the effects of T are expected to affect the APM's performance, and changes in $\mu$, density, and diffusivity in the carrier gas likely alter the radial acceleration of flow in the APM; however, further research is recommended for these aspects.

## Acknowledgments

The authors are grateful for the financial support provided by the National Science Council, Taiwan (NSC102-2221-E-008-004-MY3). This manuscript was edited by Wallace Academic Editing.

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


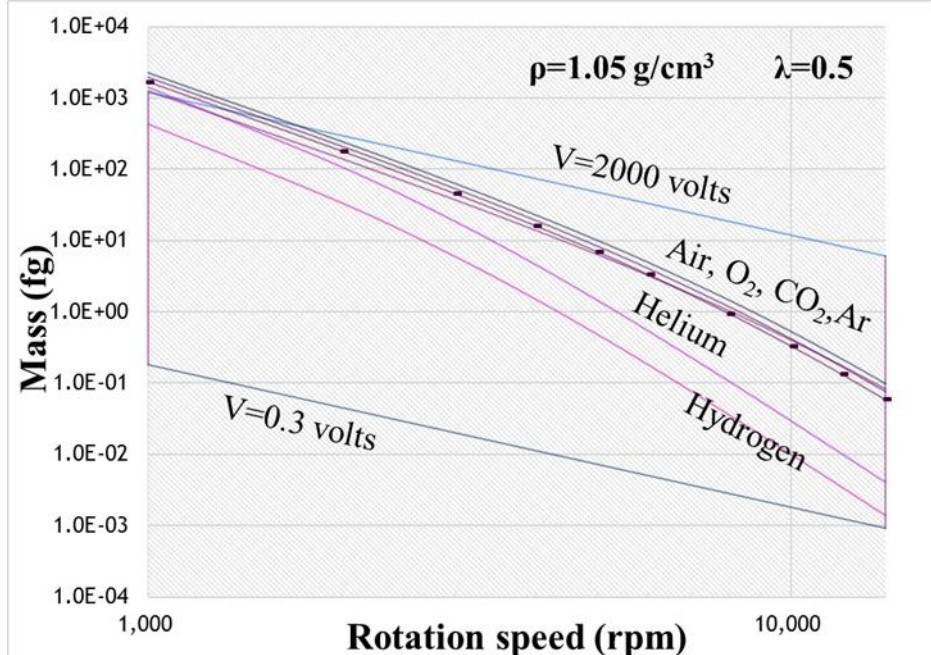

**Figure 1: Operational region of APM.**




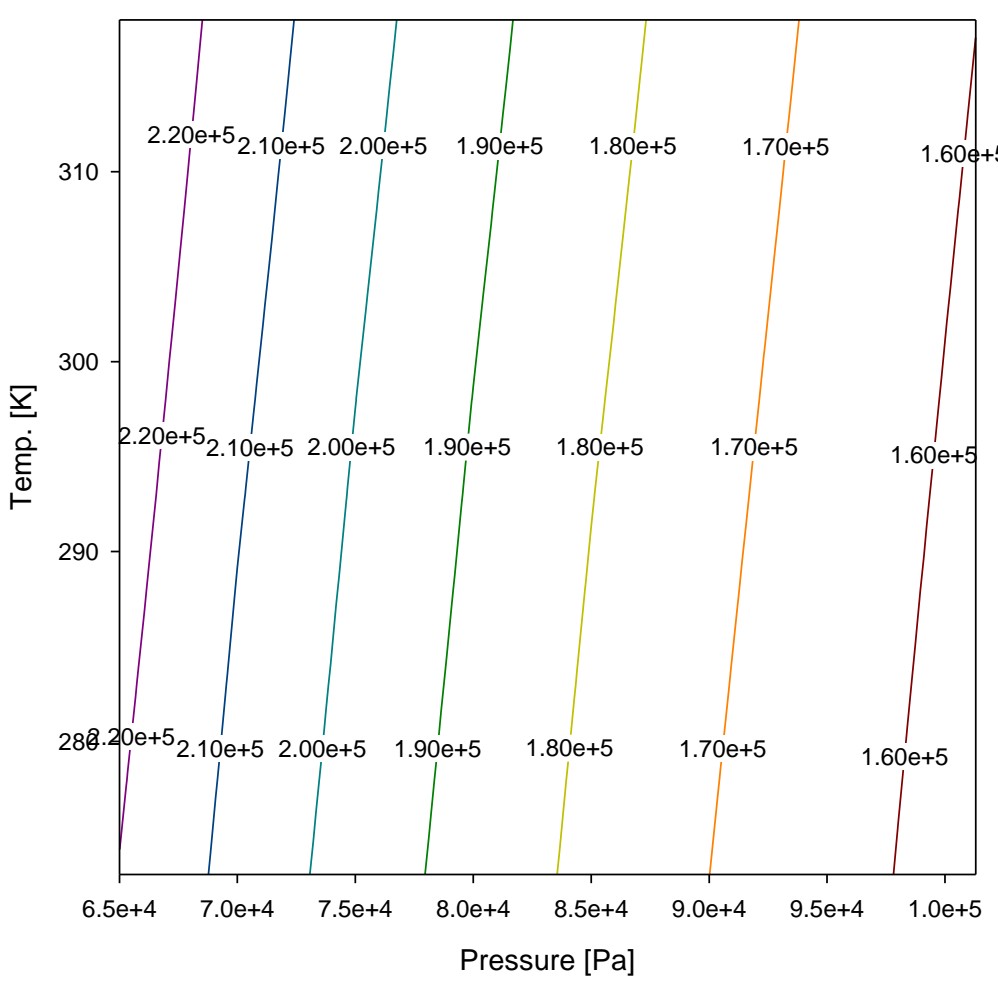

**Figure 2:** $\frac{C_c}{\mu}$ **ratio under various temperatures and pressures.**





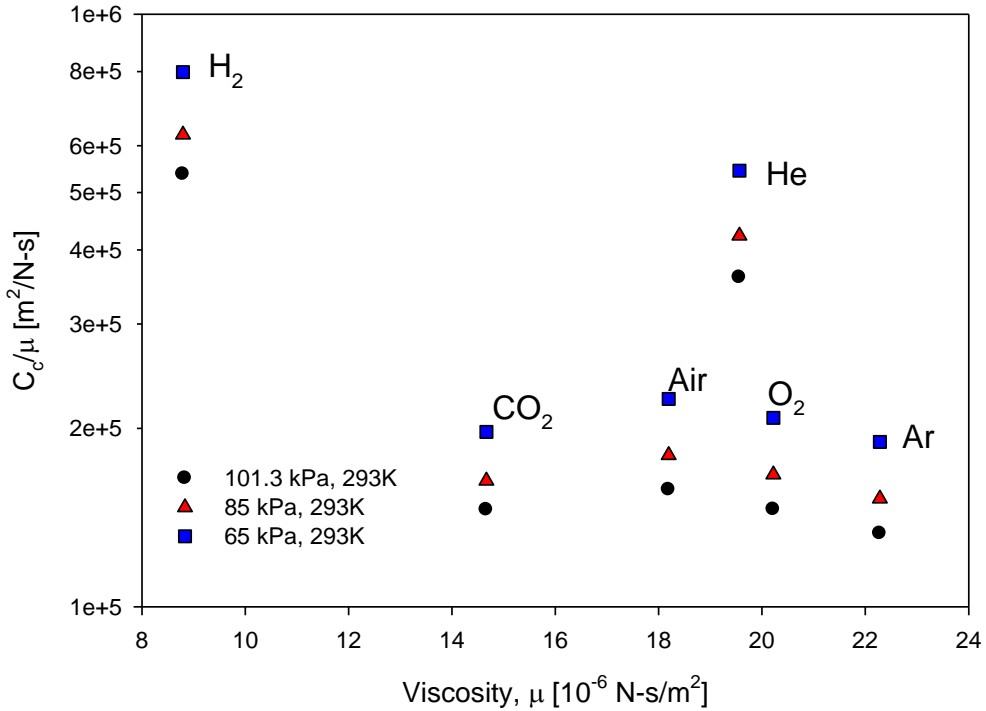

**Figure 3: Pressure effect on Cc-to-µ ratio under various carrier gases.**

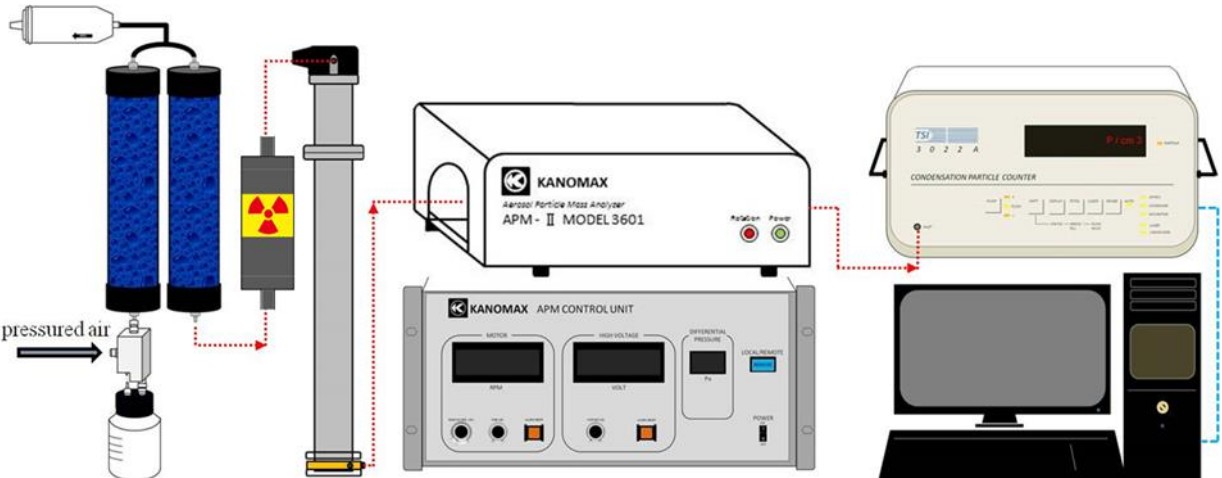

**Figure 4: Experimental DMA–APM system.**



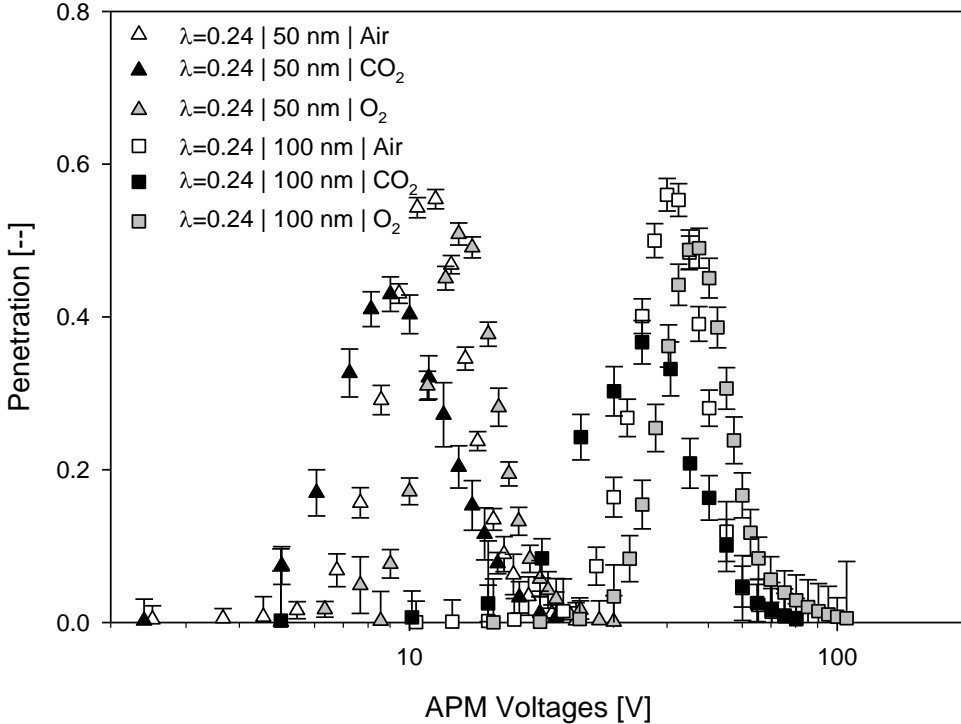

**Figure 5: APM experimental transfer function of 50- and 100-nm PSL spheres under various carrier gases.**





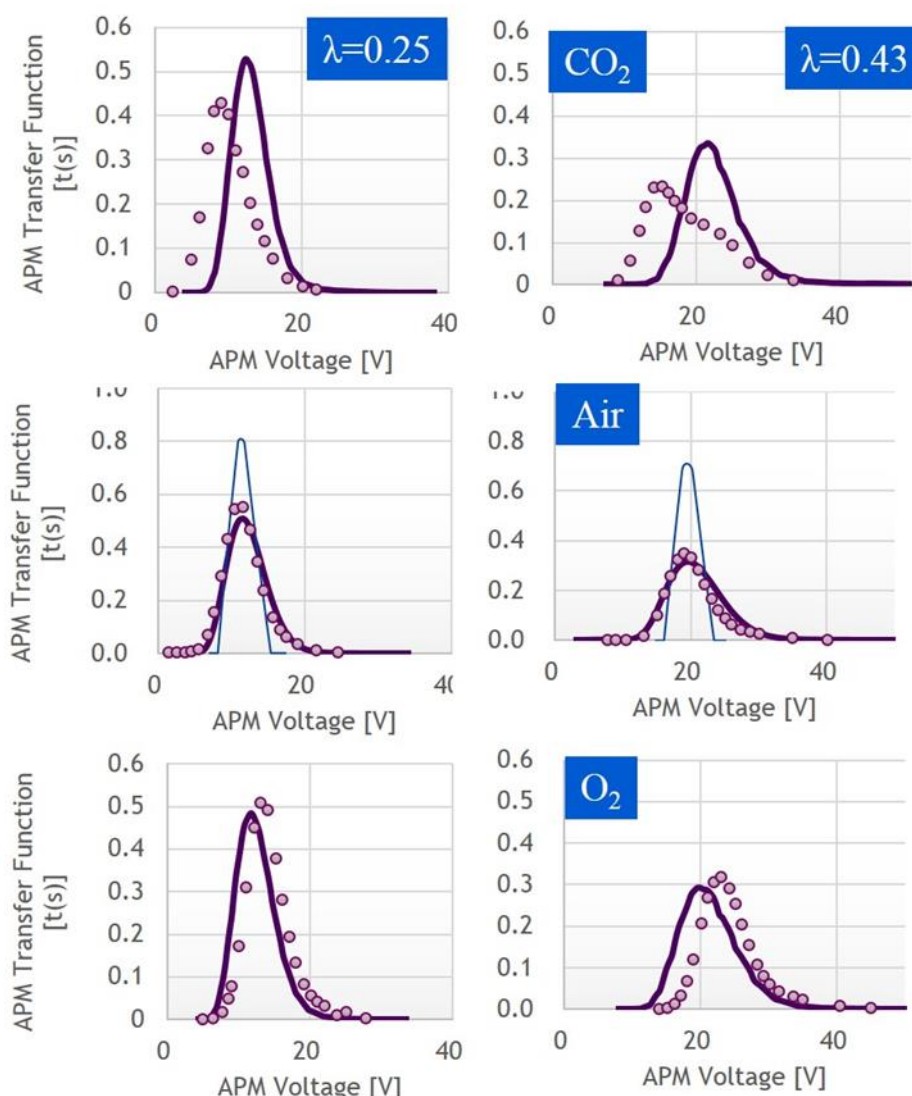

**Figure 6: APM transfer function of 50-nm PSL spheres under various carrier gases with λ = 0.25 and 0.43.**





Figure 7: Computational fluid dynamic–simulated CO₂ and O₂ flow fields at inlet and outlet regions of APM under various rotation speeds.



**Table 1: Properties of gases at normal temperature and pressure (Kulkarni et al., 2011) and corresponding APM detection limits at λ = 0.5.**

| Gas | M (g/mol) | ρ (Kg/m³) | $S_u$ (K) | $μ_r$ ($10^{-6}$ N·s/m²) | $ℓ_r$ (nm) | $d_{p,m}$* (nm) | Min. Mass ($10^{-2}$fg) | Max. Mass ($10^{-2}$fg) |
|---|---|---|---|---|---|---|---|---|
| $CO_2$ | 44.010 | 1.842 | 220.5 | 14.673 | 43.2 | 53.3-1316.9 | 8.3 | 1.3 |
| Air | 28.966 | 1.205 | 110.4 | 18.203 | 66.5 | 47.5-1448.8 | 5.9 | 1.7 |
| $O_2$ | 32.000 | 1.331 | 116.8 | 20.229 | 69.1 | 51.8-1524.8 | 7.6 | 1.9 |
| Ar | 39.948 | 1.661 | 141.4 | 22.292 | 69.4 | 56.4-1608.3 | 9.9 | 2.3 |
| $H_2$ | 2.016 | 0.090 | 66.7 | 8.799 | 123 | 13.6-921.3 | 0.14 | 0.43 |
| He | 4.003 | 0.166 | 73.8 | 19.571 | 192 | 19.5-1365.0 | 0.41 | 1.4 |

5    *$d_{p,m}$ detection limits were calculated by assuming that ρ = 1.05 g/cm³.



**Table 2: Normalized particle mass-to-charge ratio of 50-nm PSL spheres for $CO_2$, air, and $O_2$ with λ = 0.24 and 0.45.**

| PSL size | Gas | $C_C/\mu$ | Nominal λ = 0.24 | | | | | Nominal λ = 0.45 | | | | |
|---|---|---|---|---|---|---|---|---|---|---|---|---|
| | | | Exp. λ | Rotation Speed [rpm] | Peak Voltage [V] | Mass [Kg] | $\bar{S}$ | Exp. λ | Rotation Speed [rpm] | Peak Voltage [V] | Mass [Kg] | $\bar{S}$ |
| 50[*] nm | $CO_2$ | 2.35E+5 | 0.239 | 10117 | 9.6 | 5.59E–20 | 0.744 | 0.411 | 13268 | 17.1 | 5.79E–20 | 0.775 |
| | Air | 2.68E+5 | 0.244 | 9509 | 11.4 | 7.52E–20 | 1.000 | 0.419 | 12471 | 19.5 | 7.47E–20 | 1.000 |
| | $O_2$ | 2.42E+5 | 0.248 | 9955 | 13.5 | 8.12E–20 | 1.080 | 0.427 | 13056 | 23.3 | 8.14E–20 | 1.090 |
| 100[**] nm | $CO_2$ | 1.47E+5 | 0.230 | 6333 | 39.4 | 5.85E–19 | 0.966 | 0.460 | 8956 | 79.3 | 5.89E–19 | 0.985 |
| | Air | 1.56E+5 | 0.252 | 6387 | 41.5 | 6.06E–19 | 1.000 | 0.504 | 9033 | 81.8 | 5.98E–19 | 1.000 |
| | $O_2$ | 1.44E+5 | 0.250 | 6610 | 47.2 | 6.44E–19 | 1.062 | 0.499 | 9348 | 94.3 | 6.43E–19 | 1.076 |

[*]The theoretical particle mass of a 50-nm PSL sphere is 6.87E–20 kg (assuming a density of 1050 $Kg/m^3$).

[**]The theoretical particle mass of a 100-nm PSL sphere is 5.50E–19 kg (assuming a density of 1050 $Kg/m^3$).



**Table 3: Normalized particle mass-to-charge ratio of 100-nm PSL spheres for CO₂, air, and O₂ (ω is fixed at 6375.5 rpm).**

| 100 nm PSL | Peak Voltage [V] | Mass [Kg] | $\bar{S}$ |
|---|---|---|---|
| CO$_2$, $\lambda = 0.22$ | 39.4 | 5.79E–19 | 0.860 |
| Air, $\lambda = 0.25$ | 45.9 | 6.73E–19 | 1.000 |
| O$_2$, $\lambda = 0.21$ | 48.8 | 7.16E–19 | 1.064 |