# Peer review of "Effects of Temperature, Pressure, and Carrier Gases on the Performance of an Aerosol Particle Mass Analyser"

_Atmospheric Measurement Techniques, 2017_

## Referee Comment (RC1) · Anonymous Referee #1 · 24 Apr 2018

Review of "Effects of Temperature, Pressure, and Carrier Gases on the Performance of an Aerosol Particle Mass Analyser" by Hsiao et al.

The authors reported evaluations of temperature, pressure, and carrier gas on the performance of a commercially available aerosol particle mass analyser (APM). The effects of the first two parameters (temperature and pressure) were mainly evaluated through theoretical calculation of the transfer function of APM, while effects of carrier gases (air, O2 and CO2) were experimentally evaluated with DMA pre-selected 50-nm and 100-nm PSL spheres. Results suggested that the mass detection limit of particles can be as low as 10-2 fg, and can be further extended to low values with

other carrier gases such as hydrogen or with a lower operation pressure down to 80 kPa. The treatment of the theoretical calculation of the transfer function is rigorous, and experiments were well designed and performed. The writing of the manuscript is clear, and the study is within the scope of AMT. However, I have concerns, some of which appeared in the initial review, that the authors are over-interpreting their results and ignoring a few other studies of the same sort. These concerns are detailed below in Major Comments, while a few editorial suggestions are listed as Minor Comments. I suggest publication of this manuscript in AMT after the authors address them in the revised manuscript.

Major Comments:

1. While first stating the importance of the DMA-APM system in measuring atmospheric particle mass and effective density, the authors might be a bit over-emphasizing the "low" mass detection limits using hydrogen as the carrier gas and operating under 80-kPa condition. We do not have either of these often in the atmosphere (at least in the lower portion). It is good to characterize the DMA-APM system under those conditions, but it is a bit misleading (giving an impression that 10-4 fg is easily achieved in ambient measurements) to state those numbers in the abstract. And some of them were from theoretical calculation instead of direct measurement (see below in Major Comment #2).

2. While the authors stated throughout the manuscript (and the title) that temperature, pressure, and carrier gases were evaluated for the APM performance, most of them (at least for temperature and pressure) were from theoretical calculation. I suggest the authors stating this caveat explicitly in their abstract/conclusion, in order not to mislead readers that temperature and pressure were also evaluated experimentally.

3. Kuwata et al. (ES&T, 2012; AS&T, 2015) used DMA-APM to study particle (specifically SOA) density (ES&T, 2012) and developed equations to characterize the DMA-APM system (AS&T, 2015). It is highly suggested that this manuscript is put in the
context of those previous studies with comparison and discussion.

4. The authors mentioned that viscosity and density of the carrier gases might affect the classification capability of APM (page 7, line 6). While viscosity was included in their theoretical treatment, is it possible to include different densities of those carrier gases tested? These three gases (air, O2, and CO2) have quite different molecular weights too. Is that going to affect the classification capability of APM as well?

5. As the authors stated quite frequently the usefulness of using the DMA-APM system to measure effective density, what is the measured density of PSL spheres when compared to reported values?

Minor Comments:

1. Page 2, line 15: "(Ehara et al., 1996)" to "Ehara et al. (1996)".

2. Page 2, line 22: "the past decade" to "the past decades". You cited papers in as early as 2002.

3. Page 4, line 6: "defined by(Ehara et al., 1996)" should be "defined by Ehara et al. (1996).

- 4. Page 4, line 25: "low molecular weight" to "low-molecular-weight".
- 5. Page 6, line 2: "APM C\_N-V spectra". Is "C\_N" defined?
- 6. Page 6, line 27: "peak voltage is indicate" to "peak voltage is indicated".
- 7. Page 7, line 31: "by an approximate order of" to "to approximately"?

8. Page 13, Figure 5: better to separate this figure into two panels (say for 50-nm and 100-nm PSL), and join the symbol with lines. It is very difficult to tell which one is which in the current form. Also I am not sure if this Penetration-Voltage plot can be called "experimental transfer function".

9. Page 14, Figure 6: although symbols and lines were defined in the text, it would be
better to put the legends here in the figure for readers to follow easily.

10. Page 15, Figure 7: this figure has different appearance in tick labels and legends (too small to see) compared to other figures. Suggest to change those labels to a larger font size.

**AMTD**

---

## Referee Comment (RC2) · Anonymous Referee #2 · 1 May 2018

This manuscript by Hsiao et al. entitled as 'Effects of Temperature, Pressure, and Carrier Gases on the Performance of an Aerosol Particle Mass Analyser' discusses the influence of carrier gas on the operation of the APM. As far as I know, most of previous work on the APM has been focusing on operations under a normal atmospheric condition. The experimental result of the present study will help interpreting experimental data of the APM (or DMA-APM system) in the future, especially when the instrument is operated under unusual conditions. My major concern about the manuscript is that the experimental part of the study focuses on the operation of the APM under different types of gases (air, $CO_2$, and $O_2$). No experiment seems to have been conducted to investigate the influence of temperature and pressure on the APM, even though the

title mentions them. It is not clear to me when CO2 or O2 could be the major carrier gas of aerosol particles during atmospheric measurement. In that sense, I am not sure if the manuscript really fits well with the scope of the journal. I leave this question to the handling editor of the manuscript. My comments in this review focuses on scientific/technical components of the manuscript.

P5L11 Detailed information about the standard PSL particles is important for papers characterizing instruments. Please provide further detailed information about the PSL particles (e.g. manufacturer, model number, standard deviation of the distribution). Also, please provide more detailed information about the generation and desiccation processes of the PSL particles. P5L12 It was not clear to me how the DMA voltage has been set. Both the transfer function of the DMA and size-distribution of the PSL particles have relatively sharp distributions. Thus, it is important set the DMA voltage so that the center of the DMA transfer function matches with that of the size-distribution of the PSL particles. P5L21 The authors found 6% of differences in the size of PSL particles when they were measured under different types of gas. Although the authors mention that it is not significant, I do not think that the difference is small. I wonder if they have any explanations on it. P6L20 'The results revealed that particle mass was generally underestimated for cases where CO2 was used as a carrier gas. In particular, underestimation was 23%-25% for a 50-nm PSL sphere. By contrast, when O2 was used as the carrier gas, an overestimation of mass measurements was observed, with an error within 9%.' I wonder how the authors explain it. P6L25 '(convoluted with the known size distribution classified by DMA)' It might also be needed to consider the size-distribution of particles entering the DMA when they have a narrow distribution. I wonder if the authors could add any comments on it. P7L5 'Therefore, we suspect that the fluid field in the APM classification zone, also known as Taylor–Couette flow, is influenced by gas-specific properties such as $\mu$ and .' I am not sure if the APM only relies on the viscosity to rotate gas between the two rotating cylinders. There might be some kinds of internal structures to force the gas to rotate at the same angular velocity as the rotating cylinders. I suggest the authors to contact the manufacturer for

it. Figure 6: Are the y-axis of the figure the APM transfer functions, or are they the number concentration of particles measured by the CPC?

---

## Author Comment (AC1) · 21 Jun 2018

The authors would like to thank the positive feedbacks to our work and appreciate the reviewers' valuable comments for helping to significantly improve the manuscript. We agreed with most of the comments and have revised the manuscript accordingly. Following are our point-by-point response to each of the comments made by the reviewers.

Reviewer 1 Comments The authors reported evaluations of temperature, pressure, and carrier gas on the performance of a commercially available aerosol particle mass analyser (APM). The effects of the first two parameters (temperature and pressure) were

mainly evaluated through theoretical calculation of the transfer function of APM, while effects of carrier gases (air, O2 and CO2) were experimentally evaluated with DMA pre-selected 50-nm and 100-nm PSL spheres. Results suggested that the mass detection limit of particles can be as low as 10Ë̈Ę-2 fg, and can be further extended to low values with other carrier gases such as hydrogen or with a lower operation pressure down to 80 kPa. The treatment of the theoretical calculation of the transfer function is rigorous, and experiments were well designed and performed. The writing of the manuscript is clear, and the study is within the scope of AMT. However, I have concerns, some of which appeared in the initial review, that the authors are over-interpreting their results and ignoring a few other studies of the same sort. These concerns are detailed below in Major Comments, while a few editorial suggestions are listed as Minor Comments. I suggest publication of this manuscript in AMT after the authors address them in the revised manuscript.

We appreciate the positive feedbacks to our work and would revise the manuscript according to reviewer's suggestions. The abstract was rewrote to explicitly express that the effects of pressure and temperature were theoretically analysed, while the effect of carrier gas was evaluated experimentally. Using gases other than air, such as CO2 and O2, as carrier gas for APM are just trying to test whether or not the APM performance would change significantly under various conditions, which may be the case for ambient monitoring or characterizing atmospheric aerosols. For example, argon would be required as the carrier gas if the APM was used as an aerosol particle classifier coupled with inductively coupled plasma mass spectrometry (ICP-MS; in a similar manner to the DMA–ICP-MS system). Tandem APM-ICP-MS could be employed for realtime characterization of trace elements in atmospheric aerosols. In addition, the theoretical calculation and numerical simulation were conduct for explaining the experimental results.

Major Comments: 1. While first stating the importance of the DMA-APM system in measuring atmospheric particle mass and effective density, the authors might be a bit

over-emphasizing the "low" mass detection limits using hydrogen as the carrier gas and operating under 80-kPa condition. We do not have either of these often in the atmosphere (at least in the lower portion). It is good to characterize the DMA-APM system under those conditions, but it is a bit misleading (giving an impression that 10Ȩ̈Ę-4 fg is easily achieved in ambient measurements) to state those numbers in the abstract. And some of them were from theoretical calculation instead of direct measurement (see below in Major Comment #2).

Thanks for your comments. We rewrote the abstract and try to avoid over-emphasizing the detection limits. P1.L15-23: "...In this study, the effects of temperature and pressure were analysed through theoretical calculation, and the influence of varying carrier gas was experimentally evaluated. The transfer function and APM operational region were further calculated and discussed to examine their applicability. Based on the theoretical analysis of the APM's operational region, the mass detection limits are changed with the properties of carrier gases under a chosen $\lambda$ value. Moreover, the detection limit can be lowered when the pressure is reduced, which implies that performance may be affected during field study. In experimental evaluation, air, oxygen, and carbon dioxide were selected to atomize aerosols in the laboratory with the aim of evaluating the effect of gas viscosity on the APM's performance. Using monodisperse polystyrene latex (PSL) spheres with nominal diameters of 50 and 100 nm, the classification performance of the APM were slightly varied with carrier gases, while the classification accuracy were consistently within 10%."

2. While the authors stated throughout the manuscript (and the title) that temperature, pressure, and carrier gases were evaluated for the APM performance, most of them (at least for temperature and pressure) were from theoretical calculation. I suggest the authors stating this caveat explicitly in their abstract/conclusion, in order not to mislead readers that temperature and pressure were also evaluated experimentally.

Thanks for your comment. As you see in the revised manuscript, to avoid the potential misleading, the theoretical calculation and experimental evaluation were clearly defined

in abstract and main text. P1.L15-16: "...In this study, the effects of temperature and pressure were analysed through theoretical calculation, and the influence of varying carrier gas was experimentally evaluated..."

3. Kuwata et al. (ES&T, 2012; AS&T, 2015) used DMA-APM to study particle (specifically SOA) density (ES&T, 2012) and developed equations to characterize the DMA APM system (AS&T, 2015). It is highly suggested that this manuscript is put in the context of those previous studies with comparison and discussion.

As suggested by reviewer, the related references were included in the context. P2.L22-24: "...Throughout the past decades, this scheme has also been adopted extensively to determine the Df of aerosol aggregates (Lall et al., 2008; McMurry et al., 2002; Park et al., 2003; Park et al., 2004a; Park et al., 2004b; Scheckman et al., 2009) and atmospheric aerosols (Kuwata and Kondo, 2009; Kuwata et al., 2011)."

P6.L5-9: "The operation of DMA-APM is identical to Kuwata and Kondo (2009) and Kuwata et al. (2011), in which the DMA selects particles with +1 charge and predetermined mobility diameters and then subjects them to the APM. Following, the APM was set to scan across a range of voltage (V) while the number concentration (CN) of the passing particles was measured by a CPC. The peak of the CN-V distribution was subsequently inspected to determine the particle mass (m)."

P7.L5-7: "...It should be noted that, according to the work done by Kuwata (2015), even when the resolution of the DMA–APM system appears to be controlled by the APM, the particle classification by the DMA–APM at a certain operating condition still could be regulated by both DMA and APM."

P7.L26-32: "According to Kuwata's theoretical analysis of transfer function and resolution of the DMA-APM system, the common operation of constant ïĄů and varying V could not maintain the transfer function because of the range of dp,m passing the DMA (Kuwata, 2015). In such case, the transfer function may not be symmetric, and the transfer function is narrower for larger m because the dependence of ïĄňc on m. It was

then concluded that the operation of constant V and varying ïĄů, on the other hand, could better maintain the DMA-APM resolution because mïĄů2 can be constant under constant V. However, this ideal operation protocol is less employed for the DMA-APM system, mainly due to the practical impediment of quickly and accurately scanning ïĄů over a range. Therefore, the common constant ïĄů operation is investigated here."

4. The authors mentioned that viscosity and density of the carrier gases might affect the classification capability of APM (page 7, line 6). While viscosity was included in their theoretical treatment, is it possible to include different densities of those carrier gases tested? These three gases (air, O2, and CO2) have quite different molecular weights too. Is that going to affect the classification capability of APM as well?

The effect of densities of carrier gas may affect the classification performance of APM through influencing the flow field. In this study, our analytical treatment does not explore the velocity term. However, the effect of gas density was included in the numerical simulation of flow field using air, O2, and CO2 as carrier gases. The setting values are listed as below:

Density (kg/m3) Viscosity (Pa S) air 1.205 0.000018203 CO2 1.842 0.000014673 O2 1.331 0.000020229

Base on ideal gas law, the molecular weight is proportional to the density of gas under constant pressure and temperature. Therefore, we can regard the molecular weights effect as the density effect.

5. As the authors stated quite frequently the usefulness of using the DMA-APM system to measure effective density, what is the measured density of PSL spheres when compared to reported values?

The measured density for 50-nm and 100-nm PSL are 1142-1148 Kg/m3 and 1142-1156 Kg/m3, respectively. The PSL density reported by manufacturer is 1050 Kg/m3.

Minor Comments:

1. Page 2, line 15: "(Ehara et al., 1996)" to "Ehara et al. (1996)".

Corrected accordingly.

2. Page 2, line 22: "the past decade" to "the past decades". You cited papers in as early as 2002.

Revised as suggested.

3. Page 4, line 6: "defined by(Ehara et al., 1996)" should be "defined by Ehara et al. (1996).

Corrected accordingly.

4. Page 4, line 25: "low molecular weight" to "low-molecular-weight".

Revised as suggested.

5. Page 6, line 2: "APM $C\_N$-V spectra". Is "$C\_N$" defined?

The related description is added in P6.L7-8: "...the APM was set to scan across a range of voltage (V) while the number concentration (CN) of the passing particles was measured by a CPC. The peak of the CN-V distribution was subsequently inspected to determine the particle mass (m)."

6. Page 6, line 27: "peak voltage is indicate" to "peak voltage is indicated".

Revised as suggested.

7. Page 7, line 31: "by an approximate order of" to "to approximately"?

Revised as suggested.

8. Page 13, Figure 5: better to separate this figure into two panels (say for 50-nm and 100-nm PSL), and join the symbol with lines. It is very difficult to tell which one is which in the current form. Also I am not sure if this Penetration-Voltage plot can be called "experimental transfer function".

Figure revised as suggested. The Penetration-Voltage plot is revised as APM spectrum, which was used in Tajima et al. (2013).

9. Page 14, Figure 6: although symbols and lines were defined in the text, it would be better to put the legends here in the figure for readers to follow easily.

Revised as suggested.

10. Page 15, Figure 7: this figure has different appearance in tick labels and legends (too small to see) compared to other figures. Suggest to change those labels to a larger font size.

Revised as suggested.

Please also note the supplement to this comment:
https://www.atmos-meas-tech-discuss.net/amt-2017-480/amt-2017-480-AC1-supplement.pdf

**Supplement:**

[revised manuscript text omitted]

---

## Author Comment (AC2) · 21 Jun 2018

The authors would like to thank the positive feedbacks to our work and appreciate the reviewers' valuable comments for helping to significantly improve the manuscript. We agreed with most of the comments and have revised the manuscript accordingly. Following are our point-by-point response to each of the comments made by the reviewers.

Reviewer 2 Comments

This manuscript by Hsiao et al. entitled as 'Effects of Temperature, Pressure, and Carrier Gases on the Performance of an Aerosol Particle Mass Analyser' discusses the influence of carrier gas on the operation of the APM. As far as I know, most of previous work on the APM has been focusing on operations under a normal atmospheric condition. The experimental result of the present study will help interpreting experimental data of the APM (or DMA-APM system) in the future, especially when the instrument is operated under unusual conditions. My major concern about the manuscript is that the experimental part of the study focuses on the operation of the APM under different types of gases (air, $CO_2$, and $O_2$). No experiment seems to have been conducted to investigate the influence of temperature and pressure on the APM, even though the title mentions them. It is not clear to me when $CO_2$ or $O_2$ could be the major carrier gas of aerosol particles during atmospheric measurement. In that sense, I am not sure if the manuscript really fits well with the scope of the journal. I leave this question to the handling editor of the manuscript. My comments in this review focuses on scientific/technical components of the manuscript.

We appreciate these valuable comments on our work. The abstract was rewrote to explicitly express that the effects of pressure and temperature were theoretically analysed, while the effect of carrier gas was evaluated experimentally. Using gases other than air, such as $CO_2$ and $O_2$, as carrier gas for APM are just trying to test whether or not the APM performance would change significantly under various conditions, which may be the case for ambient monitoring or characterizing atmospheric aerosols. For example, argon would be required as the carrier gas if the APM was used as an aerosol particle classifier coupled with inductively coupled plasma mass spectrometry (ICP-MS; in a similar manner to the DMA–ICP-MS system). Tandem APM-ICP-MS could be employed for realtime characterization of trace elements in atmospheric aerosols. In addition, the theoretical calculation and numerical simulation were conduct for explaining the experimental results.

1. P5L11 Detailed information about the standard PSL particles is important for papers characterizing instruments. Please provide further detailed information about the PSL

particles (e.g. manufacturer, model number, standard deviation of the distribution). Also, please provide more detailed information about the generation and desiccation processes of the PSL particles. Thanks for the suggestion. The information of PSL and the generation processes were added in the revised manuscript.

P5.L11-16: "Fig. 4 depicts the experimental evaluation system. The particles were generated by an aerosol atomizer (TSI, Model 3076) and dehumidified by two desiccant dyers connected in series to remove excess water content. To experimentally evaluate the classification accuracy, 50-nm and 100-nm polystyrene latex (PSL) spheres certified by National Institute of Standards and Technology (ThermoFisher SCIENTIFIC, Cat. No. 3050A and 3100A) were used here. The mean diameters of the size distributions of the 50-nm and 100-nm PSL given by the manufacturers are 46±2 nm and 100±3 nm. These PSL particles were classified using the DMA (TSI 3081) and then delivered to the APM (Kanomax modelâĚą-3601) to determine particle mass." https://www.thermofisher.com/order/catalog/product/3050A

2. P5L12 It was not clear to me how the DMA voltage has been set. Both the transfer function of the DMA and size-distribution of the PSL particles have relatively sharp distributions. Thus, it is important set the DMA voltage so that the center of the DMA transfer function matches with that of the size-distribution of the PSL particles.

The following description were added in P6.L5-8: "...The operation of DMA-APM is identical to Kuwata and Kondo (2009) and Kuwata et al. (2011), in which the DMA selects particles with +1 charge and predetermined mobility diameters and then subjects them to the APM. Following, the APM was set to scan across a range of voltage (V) while the number concentration (CN) of the passing particles was measured by a CPC...."

3. P5L21 The authors found 6% of differences in the size of PSL particles when they were measured under different types of gas. Although the authors mention that it is not significant, I do not think that the difference is small. I wonder if they have any

explanations on it.

The differences were not considered significant because of (1) the NIST-certified PSL mean diameters (50, 100 nm) have an expanded uncertainty of ïĆś2 to ïĆś3 nm, corresponding to coefficients of variation of ïĆś3-ïĆś4.3%, (2) repeated measurements of PSLs show that the DMA has a precision of 1.7-8.9%, (3) the TSI DMA has sizing uncertainties of 3-3.5%, and (4) the 6% differences yield about 3-6 nm, which may not be of practical significances in field studies.

4. P6L20 'The results revealed that particle mass was generally underestimated for cases where CO2 was used as a carrier gas. In particular, underestimation was 23%-25% for a 50-nm PSL sphere. By contrast, when O2 was used as the carrier gas, an overestimation of mass measurements was observed, with an error within 9%.' I wonder how the authors explain it.

As shown in Table 1, the viscosity of CO2 was lower than that of air, whereas the viscosity of O2 was higher than that of air. These findings exhibit qualitative agreement with observations of under- or over-estimations of PSL spheres. Therefore, we suspect that the fluid field in the APM classification zone is influenced by gas-specific properties such as $\mu$ and . Based on the numerical simulation of the flow field and using the flow velocity of air as a reference, the velocity differences in an angular direction at the APM's inlet and outlet under various $\omega$ values are plotted in Fig. 7. The velocity was generally lower in the classification zone of the APM when CO2 was used as the carrier gas, and an increase in the distinct differences between CO2 and air with an increase in the rotation speed was observed. Therefore, a lower viscosity and higher gas density likely intensify the shear force required to create rotating flow inside the APM. Because of the lower rotating flow velocity, significant deviations were observed in the measured results under normal conditions in the case of CO2; this phenomenon is intensified with higher values of $\omega$ and is more significant for small particles, which are even more prone to influence from the flow field. (P7.L13-24)

5. P6L25 ‘(convoluted with the known size distribution classified by DMA)’ It might also be needed to consider the size-distribution of particles entering the DMA when they have a narrow distribution. I wonder if the authors could add any comments on it.

Thanks for the comments, and we have add some comments on P6.L29-31. “As reported by Lall et al. (2009) and Lall et al. (2008), the particle concentration measured as a function of APM voltage is wider than the APM transfer function even through the particle can be considered as “mono-disperse” in size. This is mainly due the spread in calibration particle sizes or the transfer function of the DMA.” Therefore, to further eliminate the spread propagated from DMA classification, the transfer function of the APM was calculated using software developed by the AIST of Japan. The transfer function predicted based on the known size distribution of the DMA outlet (convoluted with the known size distribution classified by DMA).

6. P7L5 ‘Therefore, we suspect that the fluid field in the APM classification zone, also known as Taylor–Couette flow, is influenced by gas-specific properties such as $\mu$ and .’ I am not sure if the APM only relies on the viscosity to rotate gas between the two rotating cylinders. There might be some kinds of internal structures to force the gas to rotate at the same angular velocity as the rotating cylinders. I suggest the authors to contact the manufacturer for it.

We have consulted the manufacturer, Kanomax, and confirmed that there is a partition inside electrodes to force the gas to rotate at the same angular velocity, as suggested by the reviewer. Therefore, we re-run the numerical simulation again with this internal structure and update the figures. Qualitatively, the influence of gas properties was still observed, while the quantitative effect is lessened. Therefore, the conclusion is remained unchanged and the sentence was rewrote without mentioning Tayor-Couette flow.

7. Figure 6: Are the y-axis of the figure the APM transfer functions, or are they the number concentration of particles measured by the CPC?

The y-axis of the figure should be APM transfer function.

Kuwata, M. and Kondo, Y.: Measurements of particle masses of inorganic salt particles for calibration of cloud condensation nuclei counters, Atmos. Chem. Phys., 9, 5921-5932, 2009.

Kuwata, M., Zorn, S. R., and Martin, S. T.: Using elemental ratios to predict the density of organic material composed of carbon, hydrogen, and oxygen, Environ Sci Technol, 46, 787-794, 2011.

Lall, A. A., Ma, X., Guha, S., Mulholland, G. W., and Zachariah, M. R.: Online Nanoparticle Mass Measurement by Combined Aerosol Particle Mass Analyzer and Differential Mobility Analyzer: Comparison of Theory and Measurements, Aerosol Science and Technology, 43, 1075-1083, 2009.

Lall, A. A., Rong, W., Mädler, L., and Friedlander, S. K.: Nanoparticle aggregate volume determination by electrical mobility analysis: Test of idealized aggregate theory using aerosol particle mass analyzer measurements, Journal of Aerosol Science, 39, 403-417, 2008.

Please also note the supplement to this comment:
https://www.atmos-meas-tech-discuss.net/amt-2017-480/amt-2017-480-AC2-supplement.pdf

**Supplement:**

[revised manuscript text omitted]